# Inferring bacterial transmission dynamics using deep sequencing genomic surveillance data

Madikay Senghore [1,9] ✉, Hannah Read [2,9], Priyali Oza [2], Sarah Johnson[2], Hemanoel Passarelli-Araujo[1,3], Bradford P. Taylor[1], Stephen Ashley[2], Alex Grey [2], Alanna Callendrello[1], Robyn Lee[1,4], Matthew R. Goddard [5,6], Thomas Lumley[7], William P. Hanage [1,10] & Siouxsie Wiles [2,8,10] ✉

Identifying and interrupting transmission chains is important for controlling infectious diseases. One way to identify transmission pairs – two hosts in which infection was transmitted from one to the other – is using the variation of the pathogen within each single host (within-host variation). However, the role of such variation in transmission is understudied due to a lack of experimental and clinical datasets that capture pathogen diversity in both donor and recipient hosts. In this work, we assess the utility of deep-sequenced genomic surveillance (where genomic regions are sequenced hundreds to thousands of times) using a mouse transmission model involving controlled spread of the pathogenic bacterium *Citrobacter rodentium* from infected to naïve female animals. We observe that within-host single nucleotide variants (iSNVs) are maintained over multiple transmission steps and present a model for inferring the likelihood that a given pair of sequenced samples are linked by transmission. In this work we show that, beyond the presence and absence of within-host variants, differences arising in the relative abundance of iSNVs (allelic frequency) can infer transmission pairs more precisely. Our approach further highlights the critical role bottlenecks play in reserving the within-host diversity during transmission.

The control and/or elimination of infectious diseases depend on identifying and interrupting transmission chains, particularly during acute epidemics[1–3]. While classical epidemiological techniques such as contact tracing remain integral to the epidemic control toolkit, they can be supported and strengthened by modern technological approaches[4]. For example, genomic analysis can shed light on the emergence of variants to reconstruct transmission chains in an epidemic[5–9]. The increased resolution provided by genomics has allowed epidemiologists to guide both reactive measures through retracing the transmission routes in outbreaks[6] and proactive measures after discovering environmental reservoirs[10].

Whole genome sequencing is also an essential tool to identify novel routes of host-to-host transmission[11]. Using genomics to infer transmission is limited by the diversity of the circulating pathogen

[1]Center for Communicable Disease Dynamics, Department of Epidemiology, Harvard TH Chan School of Public Health, Boston, MA, USA. [2]Bioluminescent Superbugs Lab, Department of Molecular Medicine and Pathology, University of Auckland, Auckland, New Zealand. [3]Department of Biochemistry and Immunology, Federal University of Minas Gerais, Minas Gerais, Brazil. [4]University of Toronto Dalla Lana School of Public Health, Toronto, ON, Canada. [5]School of Biological Sciences, University of Auckland, Auckland, New Zealand. [6]School of Life and Environmental Sciences, University of Lincoln, Lincoln, UK. [7]Department of Statistics, University of Auckland, Auckland, New Zealand. [8]Te Pūnaha Matatini, Centre of Research Excellence in Complex Systems, Auckland, New Zealand. [9]These authors contributed equally: Madikay Senghore, Hannah Read. [10]These authors jointly supervised this work: William P. Hanage, Siouxsie Wiles. ✉e-mail: msenghore@hsph.harvard.edu; s.wiles@auckland.ac.nz

population; two genetically similar bacteria may indicate genuine transmission between hosts, may be a result of different introductions from a third host, or arise from independent lineages with the same de novo mutational events (but with increasing numbers of unique differences this become extremely improbable). This limit is especially acute during a rapidly spreading epidemic where transmission occurs faster than fixed mutations are accumulated.

The next frontier in genomic surveillance is moving beyond single nucleotide variants (SNVs) based on consensus sequences to include further variation uncovered through deep sequencing. This allows the identification of within-host single nucleotide variants (iSNVs) present at frequencies lower than the threshold set to define a fixed mutation. The sharing of iSNVs in two closely related isolates provides extra information to aid the inference of transmission chains[7,12,13]. One such application has been the use of shared iSNVs among cases in a large US outbreak early in the Delta wave of the SARS-CoV-2 pandemic, which were used to infer transmission from a vaccinated individual and resulted in a shift in public health guidance[14].

Developing the use of iSNVs as a complement to existing outbreak investigation requires us to better understand factors such as the rates of de novo mutation supply, sequencing errors, or artefacts (which can give rise to spurious signal) and the transmission bottleneck[6,8,15] which modulates how SNVs and iSNVs are passed on from the donor to the recipient[15-17]. In practice, it is rarely possible to evaluate these because we do not directly observe transmission events.

Existing tools such as Quentin[18], TransPhylo[19] and Phyloscanner[20] can incorporate within-host diversity when inferring transmission routes, however, they have drawbacks. For example, TransPhylo uses a time-dated phylogeny that can include multiple consensus genomes from the same host. However, in an emerging outbreak, the short timescale may preclude the ability to generate a time signal. Quentin and Phyloscanner attempt to reconstruct the within-host network prior to inferring transmission chains: while Quentin implements graph and network theories, Phyloscanner subsamples sequenced reads or bam files to reconstruct subpopulations within the host. This process of reconstructing the within-host network can introduce bias that distorts the true distribution of sub-populations in the host. Moreover, Quentin assumes that transmission networks are social networks with specific properties such as power law degree distribution, small diameter, and presence of hubs. These assumptions can be erroneous in some cases such as a star like outbreak from a common source. There is a need for a standard approach that quantifies within host diversity and leverages this to infer transmission chains and uncover chains of transmission.

In this work, we take advantage of a bioluminescent derivative of the mouse enteropathogen *Citrobacter rodentium*[21,22], which we have previously used to track the bacterium within infected mice and the environment[23-25]. We establish ten transmission chains where *C. rodentium* ICC180 is controllably spread during mouse co-housing−that is, we know with certainty who infects whom at each transmission event. We deep sequence the bacteria shed from each infected animal at the point of transmission to test the hypothesis that within-host diversity can be used to offer a quantitative measure showing which isolate pairs are more likely to be linked by transmission events. Moreover, we introduce further methods to aid identification of transmission pairs by quantifying differences in the allelic frequency at iSNVs and SNV loci between sample pairs. We show that iSNVs are maintained over multiple transmission steps and that differences arising in their relative abundance can infer transmission pairs more precisely. An important component of our approach is that the inference is based solely on sequence data, without incorporating epidemiological or demographic data for context. Therefore, it can be adapted and used to complement existing epidemiologic tools.

## Results

### Observed dynamics of the transmission model in mice

Throughout this study, 220 mice were infected with/exposed to *C. rodentium* ICC180 within 10 transmission chains (Fig. 1, Supplementary Data 1). Using linear mixed models, we observed no statistically significant differences in weight losses or gains between chains or between treatments (with or without nalidixic acid supplementation) ($p$ value > 0.05, $Z$-statistic 0.61). We monitored transmission and infection dynamics by measuring viable bacterial counts (Fig. 2a–c) and bioluminescence from shed stool and by non-invasive biophotonic imaging (Fig. 2d). We observed no changes in the anatomical location of the infecting bacteria (Fig. 2d) or in disease severity, suggesting that the pathogenicity and disease dynamics of *C. rodentium* ICC180 remained unchanged over the course of the experiment. Of the animals who successfully infected their cage-mates, infections progressed as expected, with a rapid increase in bacterial numbers within the first few days, followed by a peak/plateau, and then a decline (Fig. 2a).

We also observed a large variation in the number of viable *C. rodentium* ICC180 shed by each animal around the time of transmission (Fig. 2b, Table 1), ranging from $6.83 \times 10^6$ colony forming units (CFU) g$^{-1}$ stool (W3) to $5.33 \times 10^9$ CFU g$^{-1}$ stool (W2) (Table 1). Bacterial numbers were lower in animals without nalidixic acid supplementation, but with low statistical support (Table 2). However, the transmission chains differed in both viable counts and in vivo bioluminescence data, suggesting that each chain behaved independently (Table 3). Three transmission chains (N1, N4, and W1) experienced no transmission failures, while 11 animals from the remaining chains failed to become infected (Fig. 2a, Table 1). These failures most likely reflect differences in animal grooming and coprophagic behaviour. However, it is worth noting that three of the animals came from the same cohort of 10 (M$_7$) and may have been littermates.

### Within host variants were transferred over successive transmission steps until they became fixed mutations

Our first objective was to keep track of the rate at which fixed mutations were being accrued over successive transmission steps. Fixed mutations were observed at 12 individual loci across the genome; the consensus sequences based on these 12 loci were used to reconstruct a phylogenetic tree (Fig. 3a). Out of a total of 205 isolates, 203 had an identical consensus sequence with at least one other isolate, including 126 isolates with the index strain's consensus at the basal branch on the phylogeny (Fig. 3a). Among these, 41 isolates at the base of the tree had identical consensus sequences and did not possess any within-host variants (Fig. 3a). On average, approximately one new within-host variant (iSNV) emerged with every transmission step, regardless of whether it went on to fixation, which translated to roughly 44 new within-host variants per genome per year (Fig. 3b). On average, it took 31 days for an emergent iSNV to reach fixation, either through genetic drift or selection (median 14 days, range: 7 to 126 days) (Fig. S1), which translated to 0.09 SNVs becoming fixed for every transmission step or approximately 5 SNVs per genome per year (Fig. 3c). Across the entire dataset, the mean pairwise genetic distance between isolates was 1.05 SNVs (Fig. 3d). There was no significant difference in the SNV distance of transmission pairs whether the mice were treated with nalidixic acid or not (Wilcoxon Test, $p$ value = 0.306, $W$ = 4711), however in mice fed with water the average number of new variants emerging over a transmission event was significantly higher (Wilcoxon test, $p$ value > 0.01, $W$ = 2031.5).

Typically, loci emerged as iSNVs and fixed over multiple transmission steps, while other iSNVs drifted and eventually faded away (Fig. 3e). Allelic frequency of loci changed throughout transmission chains including sweeps by subpopulations, signs of competing subpopulations and potential linkage between two mutations (Fig. 3e).

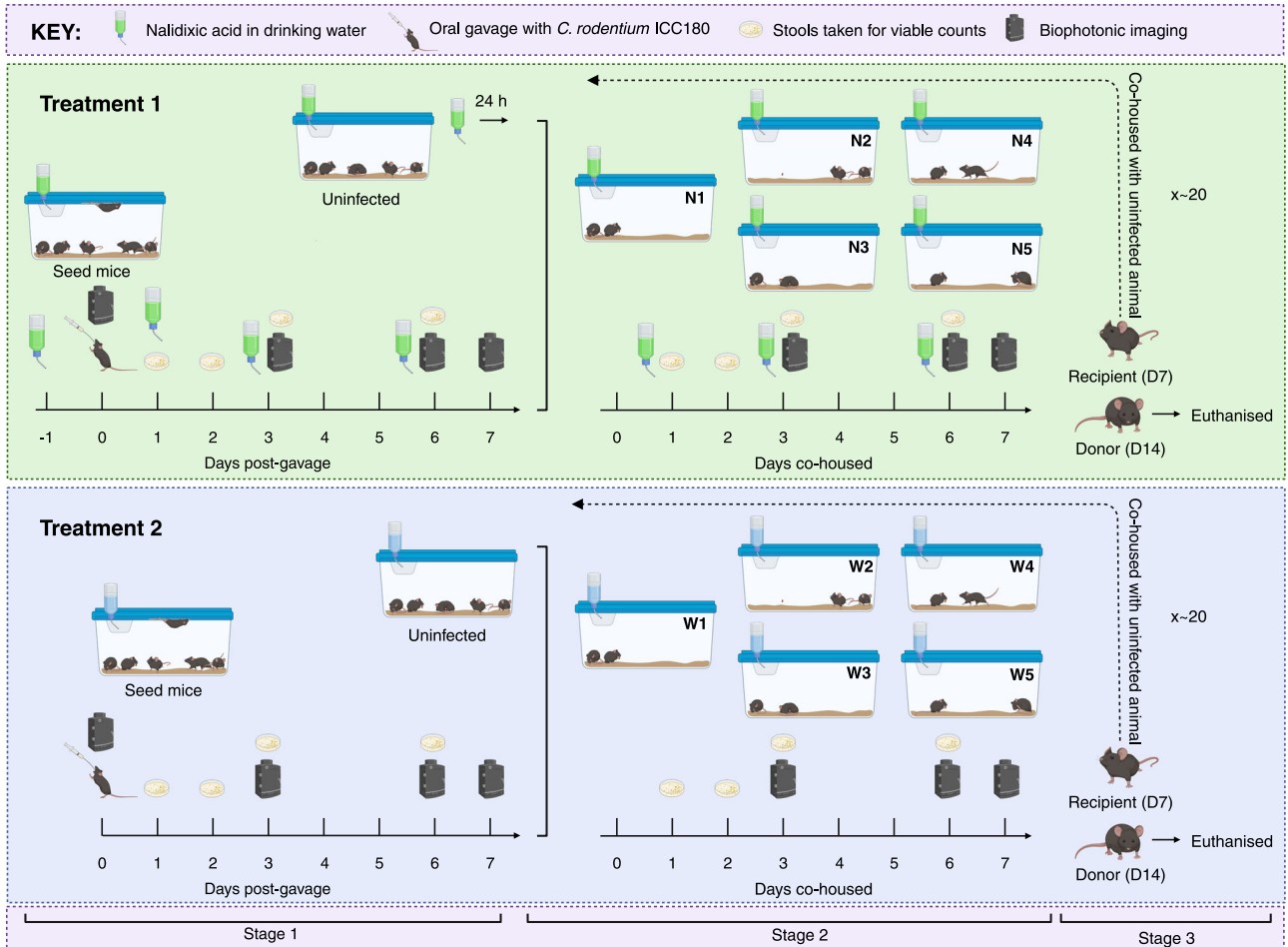

**Fig. 1 | Experimental schematic summarising the establishment of ten mouse-to-mouse *Citrobacter rodentium* ICC180 transmission chains.** Seed mice were split into two treatment groups (with nalidixic acid added to the drinking water every 2–3 days [N] or without [W]) and orally gavaged with ICC180. Seven days post-gavage, donor animals were transferred to individual cages (N1-5/W1-5) and cohoused with an uninfected cage-mate (recipient). After 7 days, the donor was humanely euthanized. The recipient then became the donor for the next step in the transmission chain by being transferred to a clean cage and cohoused with an uninfected cage-mate. This cycle was repeated until the end of the experiment. Infection and transmission dynamics were monitored by measuring luminescence and viable bacterial counts from stool samples and in vivo by biophotonic imaging.

Both stochastic and selective processes will be operating on the various bacterial lineages, but we cannot disentangle these. We classified the SNV and iSNVs and there was no statistically significant difference in the distribution of intergenic region mutations, synonymous mutations, and non-synonymous mutations between iSNVs and SNVs (Fisher's exact test *p* value = 0.98. Supplementary Data 2). Additionally, none of the iSNV or SNV loci bore mutations in two publicly available *Citrobacter* genomes: EX33 (Accession number: SAMEA782617) and DBS100 (Accession PRJNA527323 [https://www.ebi.ac.uk/ena/browser/view/CP038008]).

### Quantifying changes in the allelic frequency can improve identification of transmission pairs

To distinguish strains beyond the consensus sequence, we recorded the allelic frequency of each iSNV and SNV and quantified the magnitude of the difference at these sites. The sum of all changes in the allelic frequency was divided by the number of sites where there was a difference, and this metric was referred to as the mean change in allelic frequency (per variable site). We noted that over successive transmission steps, the mean difference in allelic frequency increased with the number of transmission steps (Fig. 4a). Moreover, linear regression showed that the mean difference in allelic frequency increases by ~0.02 units per transmission step (*p* value < 0.001, adjusted $R^2$ = 0.98) (Fig. 4b).

We also explored how predictable transmission pairs were based on the allelic frequency change and the likelihood of transmission. The likelihood was obtained by using a Bayesian framework (see Methods for details). We plotted area under curves (AUC) for three different metrics (mean allelic frequency change, total allelic frequency change, and likelihood of transmission) to test how well they performed on distinguishing transmission pairs from non-transmission pairs when the cut off was varied. All these three metrics achieved an excellent performance on distinguishing transmission pairs (AUC > 0.89) (Fig. 4c). Moreover, the mean and total changes in allelic ratio were lower in transmission pairs (Wilcoxon rank sum test, *p* value < 0.05, $W = 3536962$) (Fig. 4d, e). However, when genome artefacts were not excluded from the analysis, the model performed significantly worse, with the AUC decreasing from 0.89 to 0.56.

For each isolate, the inferred transmission likelihoods were used to identify the isolates most likely to be linked by transmission. Isolate pairs within the same transmission chain had a significantly higher transmission likelihood than isolates from different transmission chains (Wilcoxon rank sum test, *p* value < 0.05, $W = 9536430$) (Fig. 4f). In addition to investigating the potential of transmission likelihoods to discriminate transmission pairs, we also evaluated how well this metric discriminates transmission chains, by considering potential donors in transmission steps before the recipient. For example, when considering mouse $M_4$ in chain W5, we only considered mice from three prior transmission steps

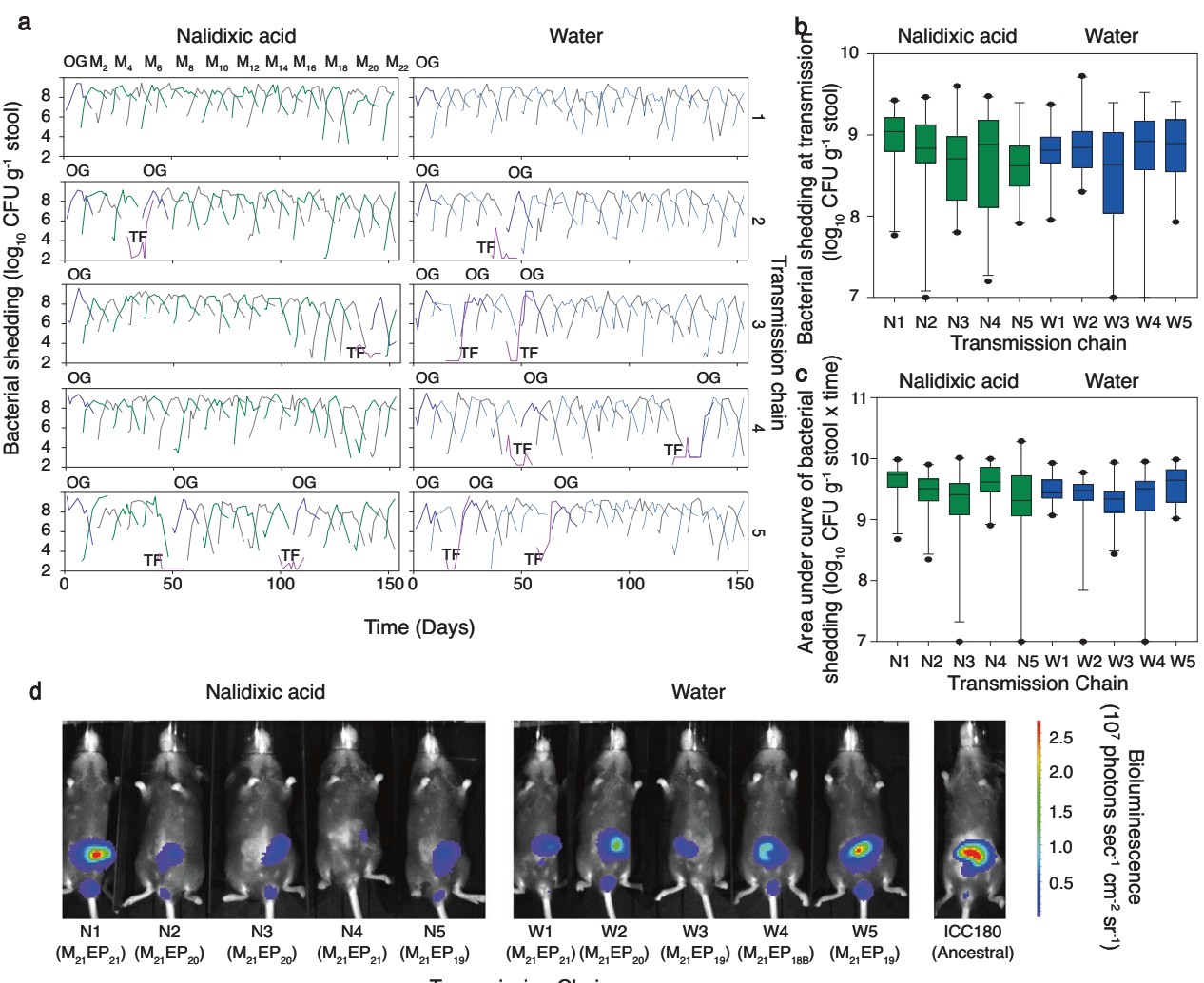

**Fig. 2 | In vivo infection and transmission dynamics of 10 mouse-to-mouse** *Citrobacter rodentium* **ICC180 transmission chains. a** Bacterial shedding in stool of each animal (as colony forming units [CFU] g$^{-1}$ stool over 14 days) with alternating colours for each subsequent transmission event (green and grey for animals in the nalidixic acid treated transmission chains; blue and grey for animals in the water transmission chains). Animals infected by oral gavage (OG) are shown in purple, while those who failed to be infected by natural transmission (TF, transmission failure) are shown in pink. **b** Box plot of bacterial shedding (as CFU g$^{-1}$ stool) at transmission summarised by transmission chain ($n = 19$-21 biologically independent mice per chain). Box plot indicates median (middle line), 25th and 75th

percentile (box), 5th and 95th percentile (whiskers) as well as outliers (single points); green boxes are from nalidixic acid treated transmission chains and blue boxes are from water transmission chains. **c** Box plot of the area under curve values of bacterial shedding (as CFU g$^{-1}$ stool $\times$ time) by transmission chain ($n = 20$ biologically independent mice per chain). Box plot indicates median (middle line), 25th and 75th percentile (box), 5th and 95th percentile (whiskers) as well as outliers (single points); green boxes are from nalidixic acid treated transmission chains and blue boxes are from water transmission chains. **d** In vivo location of *C. rodentium* ICC180 within donor mice ($M_{21}$) at the time of transmission compared to ancestral ICC180. EP, effective passage. Source data are provided as a Source Data file.

across all chains. By using only this assumption, we observed a bimodal distribution of likelihood of transmission for individuals within the same chain and a right-skewed distribution for those on different chains (Fig. 4g). Therefore, high likelihood values indicate a greater propensity to detect isolates recovered from the same chain. Moreover, by restricting to three prior transmission steps, we observed that both distributions become oppositely asymmetric, increasing the ability to correctly predict the transmission chain only with the likelihood value (Fig. 4h). Finally, the smaller the number of transmission steps (time since infection), the more left-skewed the distribution is (Fig. 4i).

We also explored the potential of using inferred likelihood to reconstruct transmission chains (Fig. 5) and to identify when the bacterium failed to transmit from mouse to mouse (Fig. 5a). The reconstruction of the network based on likelihood had an excellent performance at detecting transmission clusters (Fig. 5b). Interestingly, our model is also capable of identifying transmission at failure points

such as in chain W3 where mouse $M_4$ received the *C. rodentium* inoculum by oral gavage (see the break in the W3 strand in Fig. 5, as; highlighted in Fig. 5b).

To provide added context, we compared the effectiveness of measuring allelic frequency change versus reporting shared within-host variants (allelic frequency > 0.025) and SNV distance in predicting whether two isolates belonged to a transmission pair or the same transmission cluster. The mean change in allelic frequency outperformed SNV distance and the number of shared variants in predicting transmission pairs (AUC: 0.89 vs 0.75 and 0.80, respectively). Similarly, the mean change in allelic frequency was more effective in predicting transmission clusters (AUC: 0.82) compared to shared variants (AUC: 0.76) and SNV distance (AUC: 0.75) when the clusters were within five transmission steps. However, all three metrics had low performance in predicting whether isolates belonged to the same transmission chain (AUC < 0.75).

**Table 1 | Summary of changes in animal weight, bacterial shedding prior to housing, and transmission failures by transmission chain**

| Treatment | Transmission chain | Weight change (%) | | Viable counts prior to co-housing (CFU$^a$ g$^{-1}$ stool) | | | Transmission failures |
|---|---|---|---|---|---|---|---|
| | | At peak of infection (Median [range]) | At recovery (Median [range]) | Median (range) | Minimum | Maximum | |
| Nalidixic Acid | N1 | −0.80 (12.78) | 1.71 (12.76) | 1.10×10$^9$ (2.61×10$^9$) | 5.83×10$^7$ | 2.67×10$^9$ | None |
| | N2 | −0.96 (11.82) | 2.57 (10.93) | 6.84×10$^8$ (2.92×10$^9$) | 9.00×10$^6$ | 2.93×10$^9$ | One failure: M$_5$ |
| | N3 | 0.00 (8.61) | 3.43 (15.38) | 6.25×10$^8$ (3.94×10$^9$) | 6.33×10$^7$ | 4.00×10$^9$ | One failure: M$_{20}$ |
| | N4 | −0.29 (11.19) | 3.75 (20.40) | 7.67×10$^8$ (2.98×10$^9$) | 1.58×10$^7$ | 3.00×10$^9$ | None |
| | N5 | 1.11 (19.08) | 3.17 (12.38) | 4.17×10$^8$ (2.42×10$^9$) | 8.17×10$^7$ | 2.50×10$^9$ | Two failures: M$_7$, M$_{15}$ |
| Water | W1 | −0.27 (9.22) | 2.87 (11.82) | 6.50×10$^8$ (2.29×10$^9$) | 9.00×10$^7$ | 2.38×10$^9$ | None |
| | W2 | 0.00 (6.61) | 4.01 (13.20) | 7.00×10$^8$ (5.13×10$^9$) | 2.00×10$^8$ | 5.33×10$^9$ | One failure: M$_6$ |
| | W3 | −0.26 (7.22) | 2.79 (8.30) | 4.33×10$^8$ (2.49×10$^9$) | 6.83×10$^6$ | 2.50×10$^9$ | Two failures: M$_3$, M$_7$ |
| | W4 | 0.58 (9.40) | 3.28 (7.46) | 8.33×10$^8$ (3.32×10$^9$) | 9.83×10$^6$ | 3.33×10$^9$ | Two failures: M$_7$, M$_{18}$ |
| | W5 | 0.54 (12.77) | 3.43 (16.59) | 7.83×10$^8$ (2.50×10$^9$) | 8.50×10$^7$ | 2.58×10$^9$ | Two failures: M$_3$, M$_9$ |

Key: $^a$CFU colony forming units.

**Table 2 | Average differences in log bacterial measures (shedding and in vivo bioluminescence) per passage in a chain and between conditions (with or without nalidixic acid supplementation)**

| | | Bacterial shedding | | In vivo bioluminescence (photons s$^{-1}$) | |
|---|---|---|---|---|---|
| | | CFU g$^{-1}$ stool | RLU g$^{-1}$ stool | Abdomen | Rectum |
| Step in chain | Log ratio | −0.093 | −0.016 | −0.024 | −0.001 |
| | Standard deviation | 0.014 | 0.010 | 0.008 | 0.007 |
| Treatment | Log ratio | −0.379 | −0.379 | −0.137 | −0.264 |
| | Standard deviation | 0.317 | 0.256 | 0.179 | 0.173 |

**Table 3 | Estimated standard deviation of each measure, on the log scale, between transmission chains and between mice within a chain, and the residual standard deviations**

| | Bacterial shedding | | In vivo bioluminescence (photons s$^{-1}$) | |
|---|---|---|---|---|
| | CFU g$^{-1}$ stool | RLU g$^{-1}$ stool | Abdomen | Rectum |
| Mouse | 0.22 | 0.00 | 0.00 | 0.00 |
| Chain | 0.44 | 0.37 | 0.24 | 0.24 |
| Residual | 3.27 | 2.40 | 1.37 | 1.28 |

## Estimating bottleneck sizes and their impact on transmission inference

To study whether the observed mean changes in allelic frequency reflected actual differences arising over a transmission step, the bottleneck sizes were estimated using the beta-binomial model for all transmission steps where the donor had at least two sites with allelic frequency above the variant calling threshold (0.025). The inferred bottleneck estimates are consistent with a small bottleneck size (0–50)

(Fig. 6a). Larger bottleneck estimates were associated with wider confidence intervals and occurred among transmission pairs that had small changes in allelic frequency between pairs. The range of bottleneck estimates varied across transmission chains: chains like W5 and W4 that had larger mean changes in allelic frequency typically had smaller bottleneck sizes with shorter confidence intervals (Fig. 6b, c).

Finally, we sought to examine how the size of a transmission bottleneck affects the accuracy of inferring transmission using changes in allele frequency by simulating the transmission chains using a simplified model. Our findings indicated that the change in allelic frequency performed consistently well when the bottleneck size exceeded 10, while its effectiveness decreased as the bottleneck size decreased. Conversely, SNP distance proved to be a better predictor of transmission pairs at lower bottleneck sizes, even outperforming allelic frequency in cases where the bottleneck size was 1. However, the accuracy of SNP distance decreased significantly as the bottleneck size increased (Supplementary Fig. 3). As the rate of de novo emergence of iSNVs increased, the change in allelic frequency became less effective at lower bottleneck sizes. Conversely, under strong selection for iSNVs to become fixed, SNP distance as well as, or outperformed the change in allelic frequency in certain cases (Supplementary Fig. 3).

## Discussion

In investigations of disease outbreaks, there are three levels of linkage between cases that increase in resolution. The strictest level aims to identify exact transmission pairs, but it is also important and often valuable to distinguish between transmission chains. In fact, determining whether two isolates belong to the same outbreak is essential to the fundamental principles of outbreak investigation. It should be noted that achieving the highest level of resolution is likely feasible with genome sequencing in all cases except for those that have recently emerged (e.g., early stages of a pandemic). Our research demonstrates that even among closely related isolates, we can achieve substantial additional resolution down to the level of transmission clusters and even individual transmission links. Our method successfully established connections between isolates that originated from

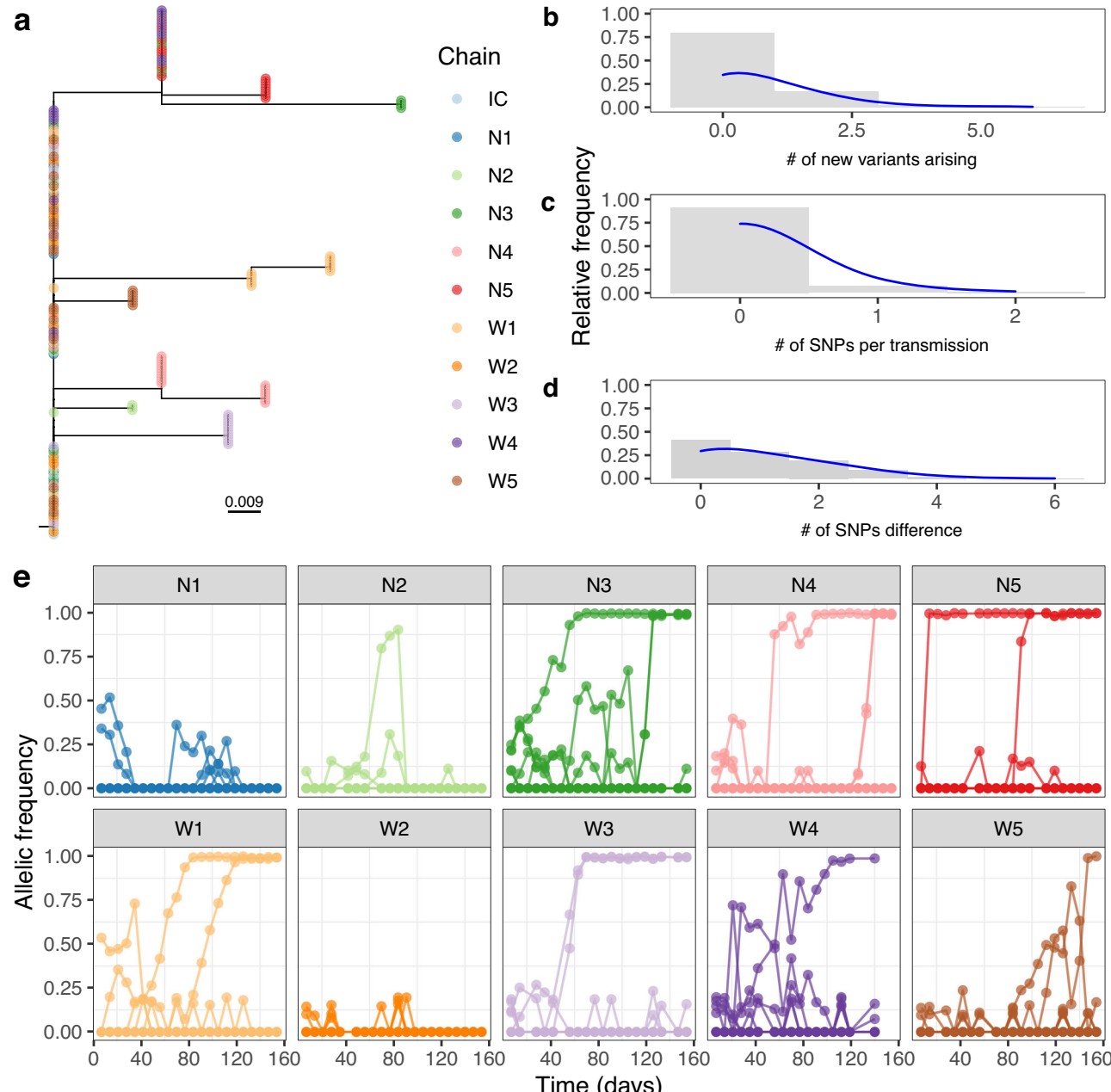

**Fig. 3 | Accumulation of diversity along transmission chains. a** Maximum likelihood phylogenetic tree based on core SNPs with tips annotated by transmission chain. The "IC" chain refers to the reference ICC180, which was used as an outgroup to root the tree. **b**–**d** Histograms showing the relative frequency for the number of iSNVs emerging in a transmission step, the number of SNVs becoming fixed during transmission, and the SNP distances across the full dataset, respectively. **e** Line graph tracking the allelic frequency of each locus throughout the ten transmission chains. Source data are provided as a Source Data file.

the same initial infected host, which can aid in the early detection of outbreaks. For instance, the SARS-CoV-2 pandemic has illustrated how large numbers of subsequent infections can result from transmission clusters originating from super-spreader events at mass gatherings[26,27]. Detecting these transmission clusters early on can significantly limit the spread of the disease. The next phase of our work will involve applying our method to datasets obtained from infections in natural and/or human populations. We will also integrate temporal, clinical, and epidemiological data with our results to further differentiate genuine transmission pairs and clusters from false positives.

When appropriately applied in the right context, our approach has the potential to aid in the early detection and interruption of transmission clusters. Even in retrospect, shared genetic variation among hosts can help confirm or refute an outbreak. For instance,

different transmission chains may have distinct implications for the allocation of public health resources compared to a single large cluster. This is especially relevant when the epidemiological characteristics differ, necessitating different interventions to (1) halt ongoing transmission chains and (2) prevent future ones in affected populations[14]. However, to effectively complement epidemiological tools/processes such as contact tracing, it is important to consider the transmission route and bottleneck sizes of the specific pathogen under investigation. Through simulations, we have determined that our method is limited by very small transmission bottlenecks e.g., five or below. The bottleneck size, which refers to the number of individual haplotypes involved in a transmission event, has an inverse relationship with the change in allelic frequency and typically follows a beta binomial distribution[16,17]. A large bottleneck size results in a small change in

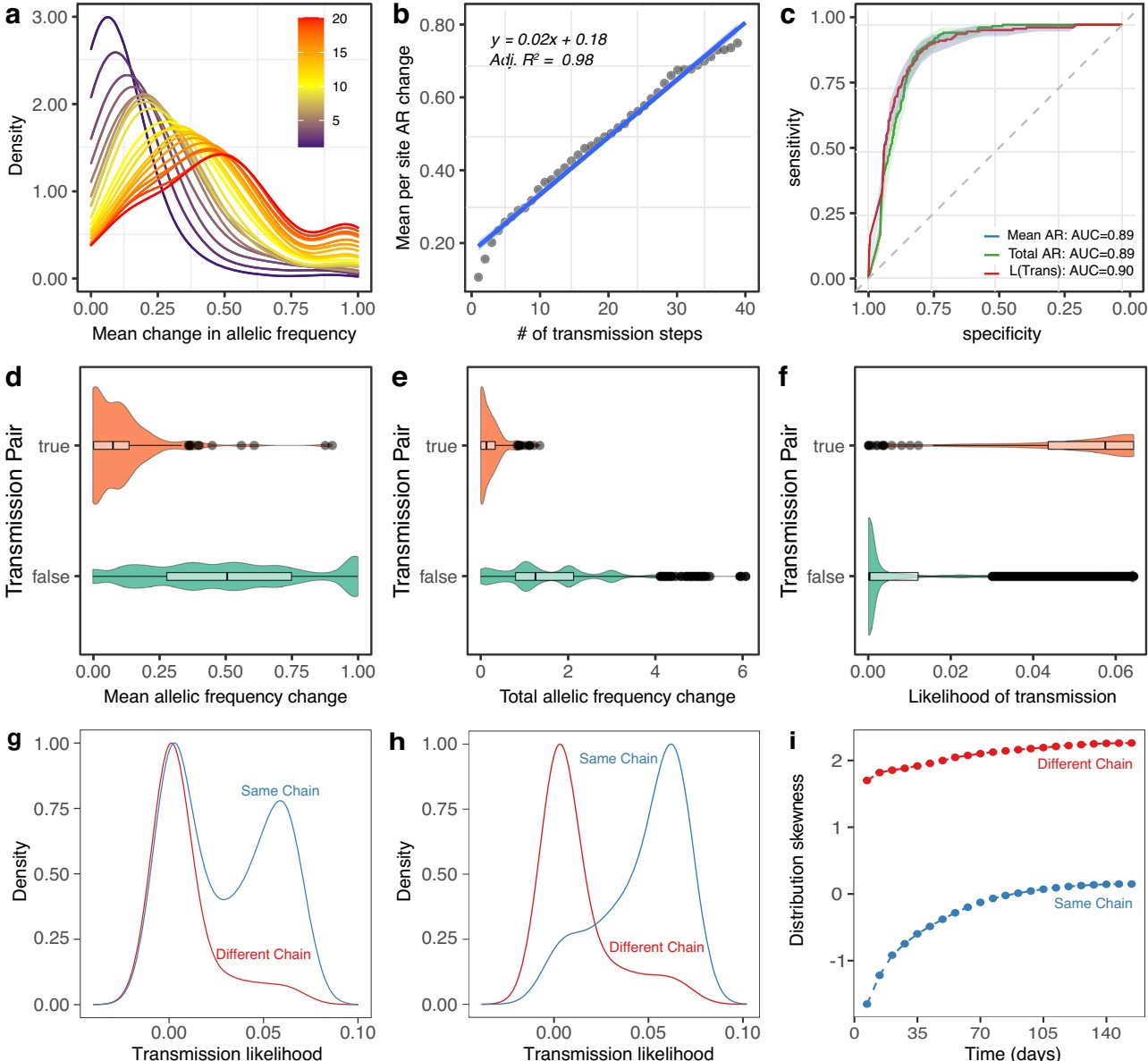

**Fig. 4 | Allelic frequency at variable loci over successive transmission steps.**
**a** Density distribution of mean allelic frequency across different transmission steps.
**b** Linear plot of average of mean allelic frequency changes, grouped by number of transmission steps. **c** Receiver operating characteristic (ROC) curves showing the performance of mean change in allelic frequency, total change in allelic frequency, and the transmission likelihood as a predictor of transmission with area under the curve (AUC). **d**–**f** Violin plots comparing the distribution of each metric in transmission pairs vs non transmission pairs; boxplots are overlayed showing the quantiles with 95% confidence intervals error bars, and dots on the graph indicate individual comparisons. **g** Distribution of transmission likelihoods between the same chain and in different chains considering donors in preceding transmission steps. **h** Distribution of transmission likelihoods considering donors at maximum three precedent transmission steps. **i** Skewness of the distribution based on incremental time/transmission steps (days). Source data are provided as a Source Data file.

allelic frequency, while a small bottleneck size can lead to a significant change in allelic frequency, indicating a smaller effective population size. Consequently, higher fluctuations in allelic frequency reduce the effectiveness of our method in inferring transmission.

Using shared variants to infer the transmission link between isolates, either directly or indirectly through intermediary transmission steps, is well supported[7,12,13,15,28–30]. However, while shared variants indicate transmission, they do not provide a quantitative measure of who is most likely to have infected whom, especially when the variant is shared across multiple individuals. A unique component of this study was that it tracked the propagation of within-host variants over multiple consecutive transmissions until some became fixed SNVs or were eliminated by a population sweep. We observed that iSNVs were maintained over multiple transmission steps, which suggests that despite being small, the bottleneck size is large enough to accommodate multiple haplotypes in a transmission event. This may explain why shared iSNVs were better at predicting whether isolates were from the same transmission chain or not.

Previous works relied on simulated datasets to quantify the role of shared variants or employed Bayesian phylodynamic models to account for unsampled hosts[7,20,31]. The challenge is that these methods are highly technical and do not always provide a quantifiable measure to inform non-bioinformatician public health officials. Our approach is unique because it quantifies the likelihood of two isolates being transmission pairs, prior to incorporating epidemiological and demographic data. Moreover, our method can differentiate

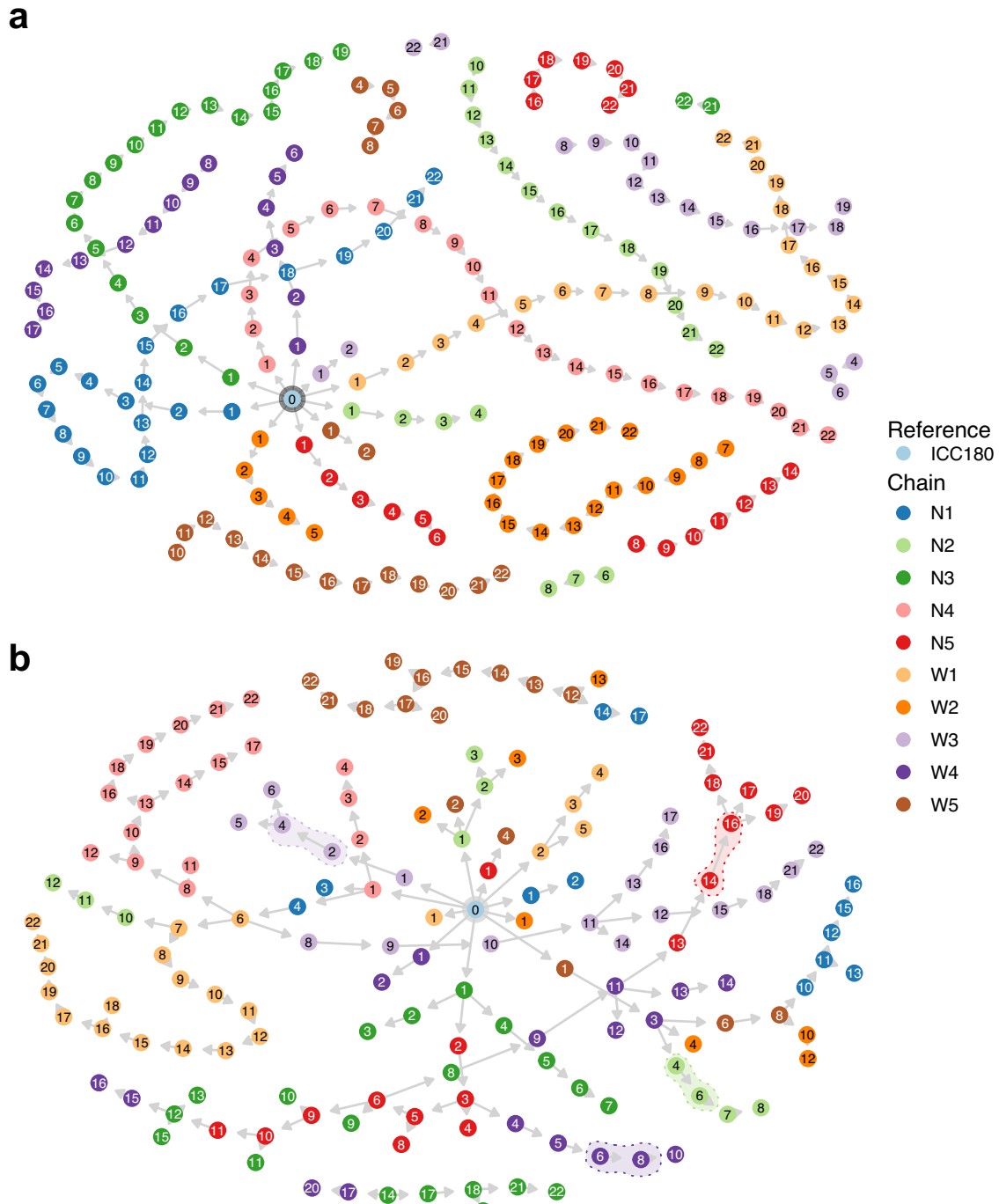

**Fig. 5 | Reconstruction of the transmission network based on the inferred likelihood. a** Ground truth of transmissions from *Citrobacter rodentium* ICC180, including transmission failures (chain breaks). Colors indicate transmission chains. **b** Transmission network from inferred likelihood values, using the time of infection as a *prior*. Transmission failures that were recognized by our method are highlighted. Source data are provided as a Source Data file.

transmission pairs among isolates that are either identical or closely related at the consensus genome level and is particularly effective at identifying isolates belonging to the same transmission chains.

This work has the following limitations. Firstly, we only sampled a single snapshot around the point of transmission. Therefore, we were unable to distinguish between diversity that arose during transmission due to a genetic bottleneck versus within host evolution during colonisation. Future work will include extending this approach to datasets where true transmission pairs and clusters are unknown. Moreover, the cost of deep sequencing remains relatively high and sequencing isn't routine in public health, however, ongoing genomic surveillance

is increasingly common and this method can be reserved for investigating suspected transmission clusters to guide rapid infection control responses (for example, SARS-CoV-2[14] or Tuberculosis[13]). Finally, this work highlights the importance of taking prudent steps to identify and remove sequencing artefacts during the analysis: we have shown that genomic artefacts can significantly decrease the specificity and sensitivity of transmission inference. We further acknowledge that this method was only tested on one pathogen in a controlled laboratory experiment, and that results from clinical and field isolates may vary.

In conclusion, we established multiple natural infection transmission chains and tracked the emergence and propagation of within-

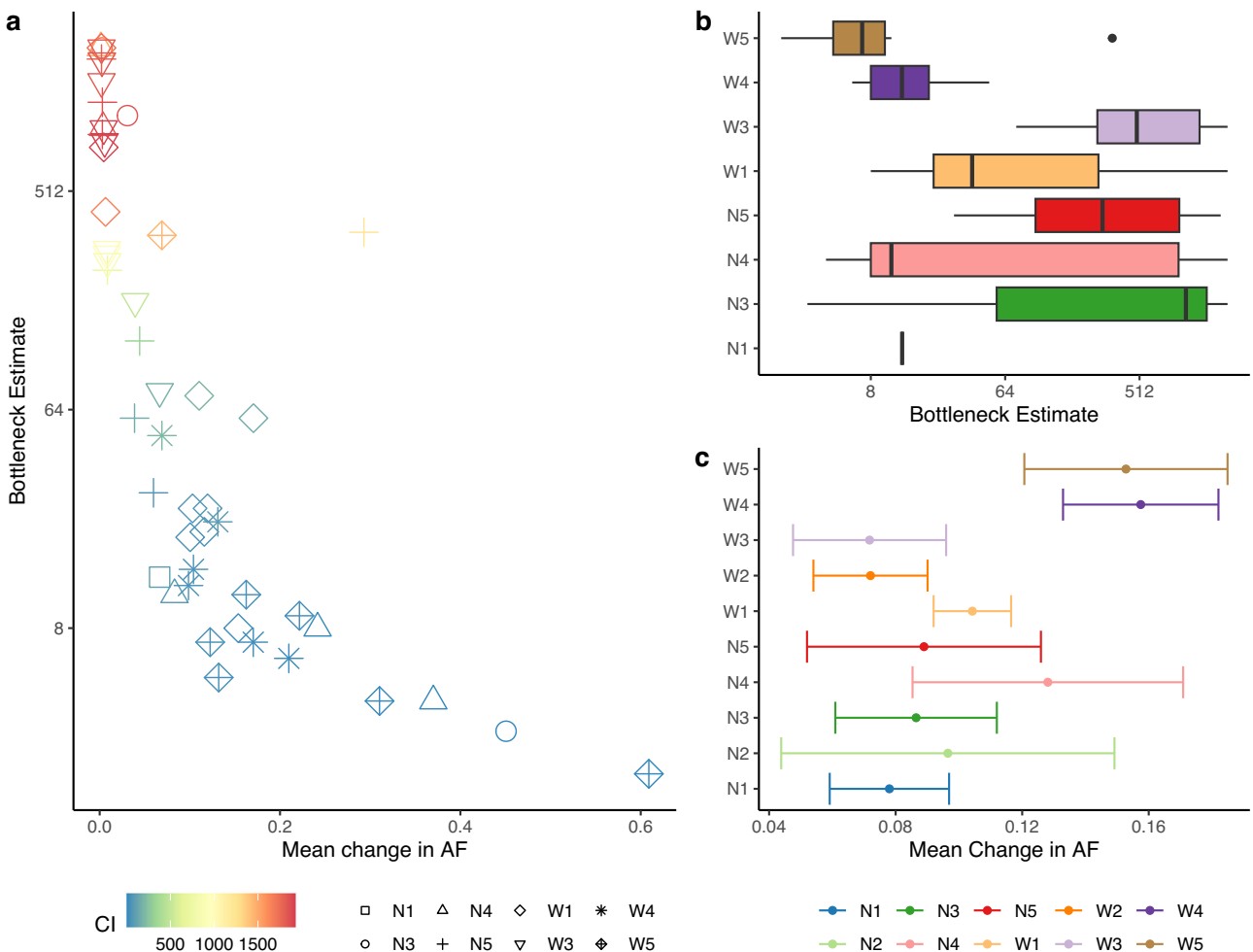

**Fig. 6 | Bottleneck estimates are consistent with a small bottleneck size (range 0-50). a** Bottleneck estimates for all transmission pairs where the donor has at least 2 sites above the variant calling threshold, shown alongside the mean change in allelic frequency (AF) corresponding to the transmission step. **b** The range of bottleneck size estimates in each transmission chain (N2 and W2 had no estimates).

Boxplots represent the quartiles and error bars represent 95% confidence intervals. **c** Range of changes in allelic frequency associated with transmission in each transmission chain. Point estimate is mean change in AF across the transmission chain and error bars represent 95% confidence intervals. Source data are provided as a Source Data file.

host variants until they became fixed SNVs or were eliminated by a population sweep. Beyond the presence and absence of within-host variants, we show that differences arising in the relative abundance of iSNVs can infer transmission clusters with high precision. Our results are an encouraging step towards higher precision contact tracing and early detection of transmission clusters. Our model can be incorporated into existing real-time sequencing frameworks and offer public health officials a quantifiable and actionable metric that can reliably infer transmission clusters.

## Methods

### Study design
Experiments were performed in accordance with the New Zealand Animal Welfare Act (1999) and institutional guidelines provided by the University of Auckland Animal Ethics Committee, which reviewed and approved these experiments under application R1003. We did not use any specific randomisation process to allocate animals to a particular transmission chain or any specific strategies to minimize any confounding factors. All authors were also aware of the group allocation.

### Animals and husbandry conditions
In this study we used female 6−7-week-old C57BL/6Elite mice provided by the Vernon Jansen Unit (University of Auckland) from specific-

pathogen-free (SPF) stocks. Male mice are more aggressive than females and if not housed together from a young age will fight, leading to stress and injury. As these experiments involved co-housing of animals from different litters, we used female mice only. Animals were housed in individually HEPA-filtered Tecniplast cages (Blue line 1284 L, Tecniplast Australia Ltd, Lane Cove, New South Wales, Australia) with sterile bedding materials (Grit-ology 1/8" corn cob [Corn-cob-ology, Mt Kuring gai, New South Wales, Australia]. Enrichment was provided in the form of EnviroDri [Biological Associates, Gladesville, New South Wales, Australia]), a mouse house (Tecniplast Australia Ltd, Lane Cove, New South Wales, Australia), and autoclaved cardboard tube. Animals were provided with free access to sterile food (Teklad global 18% protein [Biological Associates, Gladesville, New South Wales, Australia]) and autoclaved water. Conditions in the Vernon Jansen Unit are controlled at 20−24 °C, 45−65% relative humidity, and a 12-h dark-light cycle. Lights turn on at 6:30 am and off at 6:30 pm with a 30 min dawn/dusk period starting at 6 am and 6 pm, respectively.

### Establishment of transmission chains
We have published a detailed description of our methods on the protocol repository website protocols.io[32], and a schematic of the experimental design is provided in Fig. 1. Ten experimental transmission chains were established in female 6−7-week-old SPF C57BL/6Elite

mice using the bioluminescent *C. rodentium* strain ICC180. We grew *C. rodentium* ICC180 overnight at 37 °C with shaking at 200 revolutions per minute in LB-Lennox broth (Fort Richard Laboratories Ltd., Auckland, New Zealand) supplemented with kanamycin (50 µg mL$^{-1}$) (Sigma-Aldrich New Zealand Co., Auckland, New Zealand). Initially, twelve seed mice were split into two groups and orally gavaged with ICC180 (200 µL, ~10$^8$ CFU) using a straight 4 cm Instech stainless steel feeding needle (Harvard Apparatus, Holliston, MA, USA). One group of animals was given drinking water supplemented with nalidixic acid (10 µg mL$^{-1}$), which was refreshed every 2–3 days. Seven days post-gavage, the five animals with the highest ICC180 burden from each group (with nalidixic acid in the drinking water [N] or without [W]) were transferred to individual cages and designated transmission chain N1-5 or W1-5 and mouse (M)$_1$/Effective Passage (EP)$_1$. Each M$_1$/EP$_1$ animal was housed with an uninfected cage-mate, designated M$_2$/EP$_2$, for 7 days to allow for transmission to occur via grooming and coprophagia. After 7 days, the M$_1$ animals were removed and humanely euthanized; each M$_2$ animal was transferred to a clean cage and rehoused with an uninfected cage-mate, designated M$_3$/EP$_3$. We repeated this process until we reached M$_{22}$/EP$_{22}$. All animals in transmission chains N1-5 continued to receive drinking water supplemented with nalidixic acid, refreshed every 2–3 days.

## Monitoring of infection and transmission dynamics

We monitored mouse-to-mouse transmission of *C. rodentium* ICC180 by measuring luminescence and viable bacterial counts from stool samples recovered aseptically from individual animals. We homogenised stools at 0.1 g mL$^{-1}$ in PBS, measured luminescence using a luminometer (Victor X, PerkinElmer, Shelton, CT, United States) and plated them onto LB-Lennox Agar containing kanamycin (50 µg mL$^{-1}$) to determine the number of viable *C. rodentium* present per gram of stool. Stool samples were also taken from infected animals on the day they were comingled with uninfected animals, suspended 1:1 in 50% glycerol and frozen at −80 °C for genomic DNA extraction.

Where *C. rodentium* ICC180 failed to transmit between animals, we went back to the relevant frozen stool sample to produce an inoculum. For example, if *C. rodentium* failed to transmit during cohousing of animals M$_3$/EP$_3$ and M$_4$/EP$_4$, upon cohousing of M$_4$/EP$_4$ and M$_5$/EP$_5$, we orally gavaged animal M$_5$ with an inoculum produced from the frozen stools of M$_3$. Animal M$_4$ was then redesignated M$_4$/EP$_{NULL}$ and M$_5$ was redesignated M$_5$/EP$_4$ to account for the missed transmission step. Animals remained cohoused for humane reasons.

We also monitored transmission and infection dynamics using biphotonic imaging. Twice weekly we anaesthetised mice with gaseous isoflurane and measured bioluminescence using the IVIS® Kinetic imaging system (Perkin Elmer).

## Statistics & reproducibility

In this study we aimed to investigate the mutational changes that occur between consecutive infections in a transmission chain. To do this, we used 10 independent transmission chains comprising 200 independent transmission events. No statistical method was used to pre-determine sample size. However, in a pilot experiment comprising a single transmission chain, genetic changes were detectable within ten transmission events. We did not replicate the study further. Statistical analysis of infection and transmission dynamics data was carried out using the lme4 package (version 1.1–27.1)[33] in R (version 4.1). No data were excluded from the analyses. Linear mixed models were fitted to natural logarithms of bacterial load measures (CFU, bioluminescence), with fixed effects for antibiotic treatment, passage step, and time, and random effects for mouse and transmission chain. Analyses of transmission failure used logistic mixed models with a random effect for the chain.

## Whole-genome sequencing of C. rodentium ICC180 from infected animals

*C. rodentium* was obtained from frozen stool samples grown overnight at 37 °C in 10 mL LB-Lennox broth supplemented with kanamycin (50 µg ml$^{-1}$). Whole-genome DNA was extracted using a Qiagen DNeasy Blood and Tissue kit (Qiagen New Zealand Ltd, Auckland, New Zealand). The entire culture was used to capture diversity in shed stools as a surrogate for within-host diversity. Libraries were prepared using Nextera Flex (Illumina, San Diego, CA, USA). Library quality was checked by TapeStation and QPCR. Samples were sequenced at the Harvard University Bauer Core using the Illumina NovaSeq.

## Bioinformatic handling of sequence reads mapping and variant calling

We assessed read quality using FASTQC. Trimmomatic (version 0.35) was employed to remove adaptors and bases with a Phred quality score of <33. Unpaired reads and sequences less than 50 bases long were discarded. The reads were mapped to the *C. rodentium* ICC168 reference genome (Accession Number: NC_013716.1 [https://www.ncbi.nlm.nih.gov/nuccore/283783779/]) using bwa (version 0.7.17) with default parameters[34]. Samtools (version 0.1.19)[35] was used to filter unmapped reads. The GATK (version 4.0.2.1) toolkit preprocessing steps were applied to recalibrate base scores for mapped reads and perform joint variant calling[36] (Supplementary Fig. 4).

## Variant filtering and variant calling

We employed bcftools (version 1.9) to filter out variant sites with a QUAL score <100, as well as sites with indels or multiple alternative alleles. Bedtools (version 2.29.2)[37] was used to mask variants within prophage regions of the reference genome, as identified by Magaziner et al. (2019)[38]. Then, VCFtools (version 0.1.16) was used to remove consecutive variants within 100 bases window[39]. A custom Perl script was implemented to filter out sites that had fewer than 20 reads mapped. We identified 14 putative genomic artefacts where their presence in multiple transmission chains and the allelic frequency at these sites fluctuated between 0.0 and 0.3 across successive pairs of samples (Supplementary Fig. 5). Consensus SNVs were called based on allelic frequency (reads mapping to the alternative allele) of at least 90%. iSNVs were designated as loci where the allelic frequency ranged from 2.5% to 90% of reads mapping to the alternative allele. We generated an SNV alignment based on the consensus call for each genome. Associated code for variant calling is publicly available on GitHub [https://github.com/msenghore/Citrobacter_manuscript].

## Phylogenetic tree reconstruction

The core SNPs were aligned with MAFFT (version 7.467)[40]. Alignment was used as input to RAxML (version 8.2.12) to reconstruct the phylogenetic tree using the general time-reversible model and gamma correction[41]. Since we used only variable sites as input, we used ASC_GTRGAMMA to correct ascertainment bias with the Paul Lewis correction. The isolate ICC180 was used as the outgroup. One thousand bootstrap replicates were generated to assess the significance of internal nodes.

## *Pairwise comparisons and bottleneck estimates*

Statistical analysis of whole genome sequencing data was carried out in R (version 4.2.1). A pairwise SNV distance matrix based on presence and absence was calculated from all samples using Mega7 (version 7.0.267)[42]. For each pairwise comparison, we computed additional metrics to enhance our ability to distinguish transmission pairs from non-transmission pairs. First, each transmission step was taken as a single unit in time, and the sum of transmission steps separating the two isolates was recorded. For isolates in separate transmission chains, this was the sum of cumulative transmission steps from the index strain up to the isolate being queried in both chains. Then, we computed the number of

shared iSNVs and variable sites as the number of sites where both strains had an iSNV and SNV. We defined allelic frequency as the proportion of reads mapping to the alternative allele. At each variable locus, we computed the change in the allelic frequency between the two isolates. We then calculated the mean change in allelic ratio (θ), based on the number of loci where the isolates had different allelic frequencies. Finally, we employed the beta-binomial method to infer bottleneck sizes for transmission pairs based on allelic frequencies in the donor and recipient [https://github.com/weissmanlab/BB_bottleneck][16].

**Bayesian framework to infer the likelihood of transmission**
We used the mean change allelic frequency to infer the posterior probability of transmission (which we refer as likelihood of transmission). Let P(T) be the proportion of comparisons that are transmission pairs, P(θ) be the probability of observing θ and P(θ|T) the probability of observing θ in a transmission pair. The prior probability density distributions P(θ) and P(θ|T) were inferred by fitting distribution of θ for non-transmission and transmission to a truncated normal distribution using the fitdist package in R. We then calculated the posterior probability P(T|θ) of observing a transmission given the mean change in allelic frequency using Bayes theorem (Eq. (1)).

Equation 1:

$$P(\theta) = \frac{P(T) \times P(T)}{P(\theta)}$$

The obtained likelihood P(T|θ) was used to reconstruct the transmission network (Fig. 5b). At this step, we only considered transmissions that were within potential donors within three steps. We then ranked all comparisons based on transmission likelihood. The best potential donor was the one with the highest transmission likelihood. We only allowed donors in transmission steps lower than recipients.

**Reporting summary**
Further information on research design is available in the Nature Portfolio Reporting Summary linked to this article.

## Data availability
Source data provided with this paper. The raw and processed mouse infection data generated in this study have also been deposited to Figshare[43,44]. The raw sequence data generated in this study have been deposited in the NCBI short read archive database under accession code PRJNA884719. The *C. rodentium* ICC168 reference genome is publicly available in the NCBI database under the Accession Number: NC_013716.1 [https://www.ncbi.nlm.nih.gov/nuccore/283783779/]. Infection and annotated SNV data generated in this study are provided in the Supplementary Data file. Biological materials are available from the authors on request though may be subject to a materials transfer agreement. Source data are provided with this paper.

## Code availability
Associated code for variant calling and generating data tables used to create figures is publicly available on GitHub [https://github.com/msenghore/Citrobacter_manuscript].

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

## Acknowledgements

This work was supported by an R01 grant from the National Institutes of Health, awarded to W.P.H. (R01AI128344), and grants-in-aid from the University of Auckland's Faculty of Medical and Health Sciences (9802-3701152) and the Maurice Wilkins Centre for Molecular Biodiscovery (9431-48516), awarded to S.W. We acknowledge colleagues at the Centre for Communicable Dynamics, hosted at the Harvard TH Chan School of Public Health for valuable contributions and insights during the preparation of this work. We also acknowledge the assistance of ChatGPT, a language model developed by OpenAI, in providing valuable suggestions to improve the clarity of some sections of this article. We acknowledge the staff of the Vernon Jenson Unit for breeding the mice required for this study and for the care and compassion they show to all the animals in the Unit. Finally, we also acknowledge the mice used in this research, without whom this work would not have been possible.

## Author contributions

W.P.H. and S.W. conceived the study. S.W. conceived the in vivo transmission model. H.R., S.J. and S.W. carried out the in vivo experiments. M.S. and W.P.H. conceived the bioinformatic analysis. M.S., H.P.A. and B.P.T. carried out the bioinformatic analysis and modelling. M.S., H.P.A., B.P.T. and T.L. carried out the statistical analyses. S.A. prepared isolates for sequencing. S.A., A.C. and R.L. coordinated the transfer of samples. A.C. and R.L. coordinated the project management and sequencing of isolates. SW, PO, AG and MRG analyzed the in vivo data. MS and SW prepared the first draught with contributions from B..PT., H.P.A. and H.R. All authors contributed to the drafting of the manuscript.

## Competing interests

The authors declare no competing interests.
