## [Peer Review File · Nature Communications]

Reviewers' expertise:

Reviewer #1. Microbial genomics / Computational / Intra-host evolution.

Reviewer #2. Bacterial colonization / Pathogenesis / Citrobacter.

Reviewer #3. Intrahost genomics / Pathogen transmission / Computational.

Reviewers' comments:

Reviewer #1 (Remarks to the Author):

Overview: The authors present a study of controlled infection with and transmission of *C. rodentium* in laboratory mice. In particular, they focus on the ability to infer transmission chains from genomic data by leveraging within-host iSNVs and their allelic frequencies. The authors conclude that the presence/absence of iSNVs doesn't allow for the resolution of the transmission chains due to multiple occurrences of transmitted iSNVs. Hence, the proposed method considers the mean change in allelic frequency as a divergence metric. The authors show that the proposed metric can capture transmission events with higher precision than the naive method.

This work provides a great corpus of controlled transmission data with paired sequencing. The proposed method is innovative, straightforward, and sound. However, it is unclear to this reviewer how robust it is to: (a) variability in allelic frequency values caused by technical artifacts and (b) more complex evolutionary events such as indels. Additionally, it would be of high scientific value for the authors to compare their methodology to some recent transmission inference frameworks for viral data since methods for leveraging intra-host mutations have been developed and applied in various settings previously.

Overall, it would be beneficial to see a more extensive evaluation across three key directions:

(1) Impact of variability due to technical artifacts (e.g., comparing results under different variant callers or comparing results between biological replicates)

(2) A simulation experiment that includes complex evolutionary events (recombination and/or horizontal gene transfer)

(3) A direct comparison with at least one other tool that can infer transmission chains based on the intra-host variants

Major comments

- 1) In the discussion, the authors address some of the limitations of the proposed framework. However, given that the work relies on usage of iSVNs and their allelic frequencies, discussion of variation in allelic frequency due to technical reasons (e.g. sequencing depth, coverage variability, variant calling algorithm, etc.) is lacking. Technical variation can confound results and limit the resolution power of the method, and as such has to be addressed directly in the manuscript. (Also see line 145-148 comment)

- 2) When discussing the novelty of the method, authors do not mention QUENTIN, a relatively recent method for inferring transmission networks based on viral deep sequencing data and leveraging within-host mutational profiles. While QUENTIN is designed for viral data, and authors propose an approach for bacterial infections, it can still be beneficial to reference this work since it also employs a Bayesian framework in its analysis (although given a more complex model and task is likely a more computationally demanding approach).

- 3) While on a small-time scale and within a small population, the effect of within-host recombination (in the case of a viral infection) or horizontal gene transfer (in the case of a bacterial infection) can be negligible, on a scale of large outbreak or a pandemic it cannot be ignored. Given that the method relies on an averaged metric and does not provide any phylogenetic modeling, recombination and/or HGT events will likely confound the transmission inference calculations. To properly investigate and address this concern, a set of principled simulation experiments can be advised.

- 4) Lines 136-141: Some additional characterization of the mutation types can be helpful here. Are the iSNVs that fix in a population different from the ones that don't based on the amino acid change impact? Do some of the iSNVs that fix appear in other strains of *C. rodentium* found in GenBank? While I agree with authors that fully disentangling stochastic and selective components of bacterial evolution is challenging, additional analyses can help gain some partial understanding.

- 5) Lines 145-148: If I understand the definition of the mean change in allelic frequency (AF) correctly, then the scenario in which a single locus has a large AF difference will result in the same distance as multiple loci having small variations in AF. This can prove problematic in cases where due to technical variation, two isolates can have minor differences in AF across multiple loci.

Reviewer #2 (Remarks to the Author):

The paper titled “Beyond consensus sequence: a quantitative scheme for inferring transmission using deep sequencing in a bacterial transmission model” provides a meaningful way to enhance contact tracing methods using *C. rodentium* as a model pathogen. The paper is well-written and clearly addresses the strengths and weaknesses of the proposed scheme. I have listed a few comments that I believe can improve the paper.

1. Although the authors utilize the antibiotic pre-treated model of Nalidixic acid in conjunction with untreated, they haven't clearly stated the difference in *C. rodentium* strain adaptation in the gut (with and w/o Nal) and if that contributed to some of the SNVs. Can the authors discuss this in the context of their conclusion?
2. The authors are advised to increase the font size for Figure 2 axes. Its currently illegible.
3. Figure legend 2A – Please correct CFU/g
4. Line 107. Is the text not calling the correct figure?

Reviewer #3 (Remarks to the Author):

Senghore et al describe a transmission experiment, sequencing *C. rodentium* as it is passaged through a number of independent lines of mice. Alongside this they present a mathematical method to identify transmission pairs from genome sequence data. The focus of the work as presented is on the method, and its potential general applicability for purposes of contact tracing.

My view is that the experiment provides a nice dataset for the study of viral transmission in cases such as this, but that the mode of transmission described, whereby mice eat each others poop, is unrepresentative of the transmission dynamics of a large number of pathogens for which the use of genomic data for contact tracing would be of interest. Although the method works well as applied to this experiment, there are strong reasons to believe it would not improve on existing methods when applied to other pathogens. For this reason the claims made around contact tracing appear overblown.

The experiment appears to be carried out well and provides a nice dataset. To the extent that the data might be of use to other researchers it would be valuable if the data were deposited upon publication into a public repository such as the Sequence Read Archive. Given that *C. rodentium* is spread via fecal-oral transmission, the bottleneck sizes observed at transmission are generally large.

I was not 100% clear about the method used for processing sequence data, specifically whether the authors mean to cite the allelic intensity ratio θ as used e.g. by Staaf et al., BMC Bioinformatics, 2008; a reference or equation would be valuable at this point. What is clear is that the allele frequencies measured during the experiment were converted into a summary statistic, representing the amount that allele frequencies change across transmission. To first approximation, the change in an allele frequency at transmission is a function of the binomial distribution, with variance dependent upon the frequency p and the bottleneck size N . Given large N , small changes will be observed in frequencies upon transmission, increasing in a roughly linear fashion across multiple transmissions as observed in Figure 4B. The successful inference of who infected who depends upon this relationship, with small changes in allele frequency being more likely in cases of transmission than across more distant relationships (hence Figure 4D).

The problem with the method as applied to other situations is that the majority of studies looking at infectious disease transmission in humans find bottlenecks that involve close to one virus particle; this is true for influenza (McCrone et al., eLife, 2018), SARS-CoV-2 (Lythgoe, Science, 2021), and HIV (Carson et al, Science 2014): The transmission dynamics that lead to the success of the method in this case do not apply. As such, unless the contact tracer of the abstract is working on an outbreak of fecal-oral transmission in mice, it is unclear that the method would prove so valuable. The authors may have specific applications in mind, but without further clarification the claims of general applicability are not justified.

Minor points:

Equations 1 and 2 were not displayed properly in the manuscript I received. For example I think that equation 1 should have $P(T | \theta)$ on the right hand side, not simply $P(\theta)$. I think this is just a formatting error?

Not all of the data shown in Figures 4D-F seems to be appropriate for a box plot. In particular, in Figure 4F it looks as though the 'false' data are bimodal: there are so many outliers that few conclusions can be drawn from what is shown.

Line 234: “the bottleneck size is greater than or comparable to the amount of within-host diversity” -
Please clarify: in numerical terms the two statistics are measured using different units.

Reviewers' comments	Response to reviewer comments
Reviewer #1 (Remarks to the Author):	
Overview: The authors present a study of controlled infection with and transmission of C. rodentium in laboratory mice. In particular, they focus on the ability to infer transmission chains from genomic data by leveraging within-host iSNVs and their allelic frequencies. The authors conclude that the presence/absence of iSNVs doesn't allow for the resolution of the transmission chains due to multiple occurrences of transmitted iSNVs. Hence, the proposed method considers the mean change in allelic frequency as a divergence metric. The authors show that the proposed metric can capture transmission events with higher precision than the naive method. This work provides a great corpus of controlled transmission data with paired sequencing. The proposed method is innovative, straightforward, and sound. However, it is unclear to this reviewer how robust it is to: (a) variability in allelic frequency values caused by technical artifacts and (b) more complex evolutionary events such as indels. Additionally, it would be of high scientific value for the authors to compare their methodology to some recent transmission inference frameworks for viral data since methods for leveraging intra-host mutations have been developed and applied in various settings previously. Overall, it would be beneficial to see a more extensive evaluation across three key directions:  (1) Impact of variability due to technical artifacts (e.g., comparing results under different variant callers or comparing results between biological replicates) (2) A simulation experiment that includes complex evolutionary events (recombination and/or horizontal gene transfer) (3) A direct comparison with at least one other tool that can infer transmission chains based on the intra-host variants 	
Major comments	
1) In the discussion, the authors address some of the limitations of the proposed framework. However, given that the work relies on usage of iSVNs and their allelic frequencies, discussion of variation in allelic frequency due to technical reasons (e.g. sequencing depth, coverage variability, variant calling algorithm, etc.) is lacking. Technical variation can confound results and limit the resolution power of the method, and as such has to be addressed directly in the manuscript. (Also see line 145-148 comment)	In the final step of our variant calling pipeline, we identified putative sequencing artefacts that we labelled as noisy sites (see lines 353-355). To demonstrate the impact of genomic artefacts on the sensitivity and specificity of our method, we re-ran the transmission inference pipeline without excluding the 14 genomic loci that were identified as genome artefacts. The inclusion of genomic artefacts significantly decreased the ability of our model to distinguish transmission curves. The AUC of the ROC curve decreased from 0.89 to 0.56. This is now mentioned in the results and highlighted in the discussion.

2) When discussing the novelty of the method, authors do not mention QUINTIN, a relatively recent method for inferring transmission networks based on viral deep sequencing data and leveraging within-host mutational profiles. While QUINTIN is designed for viral data, and authors propose an approach for bacterial infections, it can still be beneficial to reference this work since it also employs a Bayesian framework in its analysis (although given a more complex model and task is likely a more computationally demanding approach).	"The reviewers reasonably requests comparing our method to established methods, such as QUINTIN. While we agree such comparisons would be valuable, we disagree that comparing to QUINTIN would be fruitful given the assumptions that that algorithm makes on the structure of the epidemic. Namely, from the QUINTIN manuscript "[Q]UINTIN uses the fact that generally virus transmission networks are social networks with a specific properties such as power law degree distribution, small diameter and presence of hubs". The transmission network within our experiment is star-like, far from a power-law degree distribution. This incorrect assumption will bias model results rendering the comparison fraught. As for other comparisons, we are unaware of any widely accepted methods beyond QUINTIN, and indeed, a goal of this manuscript is to propose our method as an adoptable method."
3) While on a small-time scale and within a small population, the effect of within-host recombination (in the case of a viral infection) or horizontal gene transfer (in the case of a bacterial infection) can be negligible, on a scale of large outbreak or a pandemic it cannot be ignored. Given that the method relies on an averaged metric and does not provide any phylogenetic modeling, recombination and/or HGT events will likely confound the transmission inference calculations. To properly investigate and address this concern, a set of principled simulation experiments can be advised. –	The reviewer raises an important point, regarding the role of horizontal gene transfer in evolution of an outbreak, especially a large-scale outbreak. While we recognize the importance of these factors, our pipeline is relying on a standard bacterial phylogenetic pipeline, which if carefully curated should not be influenced by horizontal gene transfer and homologous recombination. Bacterial phylogenies are based on the core genome, which is by definition devoid of horizontally acquired elements. However, it is possible that the gain and loss of genes might produce spurious signal in the flanking regions of the core genome, as a result of assembly errors. In the present MS, we have explicitly considered the potential impact of such artefacts on the sensitivity and specificity. And the issue has been previously shown to be empirically addressed by generation of a new closely related reference genome (Lee et al eLife 2020). Similarly, when analyzing samples from a broader outbreak, it is important to identify and remove regions under homologous recombination. The presence of homologous recombination within the core genome might obscure accurate phylogenies (See Didelot et. al Trends Microbial, 2010; Wilson MBio, 2014; Croucher et al Nucleic Acids Res, 2015; Croucher et al Science, 2011; Didelot et al

	Nucleic Acids Res, 2018), but it would not completely remove polymorphisms such as iSNVs, and so would not affect this method. Applying this approach as a standard for viral outbreaks requires careful thought. We must consider that viral genomes are significantly smaller, and have variable mutation rates. Moreover, viruses may employ different routes of transmission that elicit variable bottleneck sizes. Such a model is beyond the scope of this paper as we have presented a straightforward method for quantifying within host variation, which we hope can be adapted to provide added resolution in resolving transmission routes.
4) Lines 136-141: Some additional characterization of the mutation types can be helpful here. Are the iSNVs that fix in a population different from the ones that don't based on the amino acid change impact? Do some of the iSNVs that fix appear in other strains of C. rodentium found in GenBank? While I agree with authors that fully disentangling stochastic and selective components of bacterial evolution is challenging, additional analyses can help gain some partial understanding. –	We classified the effects of SNV and iSNVs and found that there was no statistically difference in the distribution of intergenic region mutations, synonymous mutations and non-synonymous mutations between iSNVs and SNVs (Appendix 1, SNP Effect table). We used the ICC168 strain as the reference genome, this strain is recognized as being representative of the Citrobacter rodentium species. Given the paucity of Citrobacter rodentium reference genomes in the public domain we were able to compare our dataset with only two Citrobacter strains EX33 (Accession number: PRJEB45982. Petty et al Plos Pathogens 2011) and DBS100 (Accession PRJNA527323, Popov et al Announcement 2019). None of the sites identified as SNVs or iSNVs in our dataset corresponded to mutations in either EX33 or DBS100.
5) Lines 145-148: If I understand the definition of the mean change in allelic frequency (AF) correctly, then the scenario in which a single locus has a large AF difference will result in the same distance as multiple loci having small variations in AF. This can prove problematic in cases where due to technical variation, two isolates can have minor differences in AF across multiple loci.	In Figure 3, we show that mutations drift to fixation at variable rates (also see supplementary figure 3B). Despite these variations, when we average out the changes in AF, we are able to substantially improve the resolution between pairs of isolates, this is one of the remarkable things about this method. It provides resolution where traditional approaches based on fixed mutation would not offer discrimination. The reviewer also makes a useful comment on technical variation, this is a problem that is inherently ubiquitous to genomic analyses and bioinformaticians are continuously improving pipelines to minimize such noise artefacts. Like all bioinformatics pipelines,

	the user must make a concerted effort to minimize the impact of spurious genomic loci. In our methods section we have clearly laid out all the steps we have taken to minimize noise, and these steps can be replicated by our colleagues who wish to replicate/ adapt our approach.
Reviewer #2 (Remarks to the Author):	
The paper titled “Beyond consensus sequence: a quantitative scheme for inferring transmission using deep sequencing in a bacterial transmission model” provides a meaningful way to enhance contact tracing methods using C. rodentium as a model pathogen. The paper is well-written and clearly addresses the strengths and weaknesses of the proposed scheme. I have listed a few comments that I believe can improve the paper.	
1. Although the authors utilize the antibiotic pre-treated model of Nalidixic acid in conjunction with untreated, they haven’t clearly stated the difference in C. rodentium strain adaptation in the gut (with and w/o Nal) and if that contributed to some of the SNVs. Can the authors discuss this in the context of their conclusion? –	There was no significant difference in the SNV distance of transmission pairs whether the mice were treated with nalidixic acid or not (Wilcoxon Test, p-value = 0.306), however in mice fed with water the average number of new variants emerging over a transmission event was significantly higher that (Wilcoxon test, p-value > 0.01).
2. The authors are advised to increase the font size for Figure 2 axes. Its currently illegible.	
3. Figure legend 2A – Please correct CFU/g	
4. Line 107. Is the text not calling the correct figure?	Corrected.
Reviewer #3 (Remarks to the Author): More discussion on bottleneck size and within host variation	
Senghore et al describe a transmission experiment, sequencing C. rodentium as it is passaged through a number of independent lines of mice. Alongside this they present a mathematical method to identify transmission pairs from genome sequence data. The focus of the work as presented is on the method, and its potential general applicability for purposes of contact tracing. My view is that the experiment provides a nice dataset for the study of viral transmission in cases such as this, but that the mode of transmission described, whereby mice eat each other’s poop, is unrepresentative of the transmission dynamics of a large number of pathogens for which the use of genomic data for contact tracing would be of interest. Although the method works well as applied to	This comment is well received, and the discussion has been modified accordingly to reduce the emphasis on contact tracing. However, there is strong evidence that our applying our method is not restricted to an experimental set up involving fecal oral route of transmission. Below we provide rebuttals, and where appropriate concurrence to the statements made by reviewer 3.

this experiment, there are strong reasons to believe it would not improve on existing methods when applied to other pathogens. For this reason, the claims made around contact tracing appear overblown.	
The experiment appears to be carried out well and provides a nice dataset. To the extent that the data might be of use to other researchers it would be valuable if the data were deposited upon publication into a public repository such as the Sequence Read Archive. Given that C. rodentium is spread via fecal-oral transmission, the bottleneck sizes observed at transmission are generally large.	The sequences for this work have been deposited in the NCBI sequence archives and are publicly available under the project accession number PRJNA884719.
I was not 100% clear about the method used for processing sequence data, specifically whether the authors mean to cite the allelic intensity ratio θ as used e.g. by Staaf et al., BMC Bioinformatics, 2008; a reference or equation would be valuable at this point. What is clear is that the allele frequencies measured during the experiment were converted into a summary statistic, representing the amount that allele frequencies change across transmission. To first approximation, the change in an allele frequency at transmission is a function of the binomial distribution, with variance dependent upon the frequency p and the bottleneck size N. Given large N, small changes will be observed in frequencies upon transmission, increasing in a roughly linear fashion across multiple transmissions as observed in Figure 4B. The successful inference of who infected who depends upon this relationship, with small changes in allele frequency being more likely in cases of transmission than across more distant relationships (hence Figure 4D).	Reference to θ in our manuscript are not associated with the paper by Staaf et al. We define θ in the methods section: lines 376-380 read “We defined allelic frequency as the proportion of reads mapping to the alternative allele. At each variable locus, we computed the change in the allelic frequency between the two isolates. We then calculated the mean change in allelic ratio (θ), based on the number of loci where the isolates had different allelic frequencies.”

The problem with the method as applied to other situations is that the majority of studies looking at infectious disease transmission in humans find bottlenecks that involve close to one virus particle; this is true for influenza (McCrone et al., *eLife*, 2018), SARS-CoV-2 (Lythgoe, *Science*, 2021), and HIV (Carlson et al, *Science* 2014): The transmission dynamics that lead to the success of the method in this case do not apply. As such, unless the contact tracer of the abstract is working on an outbreak of fecal-oral transmission in mice, it is unclear that the method would prove so valuable. The authors may have specific applications in mind, but without further clarification the claims of general applicability are not justified.

We agree that a more nuanced discussion of the limitations of the study is warranted, particularly in the context of contact tracing, and the discussion has been updated accordingly. Nonetheless, recent advances in leveraging within host diversity to inform transmission chains provide empirical support for the approach. Notably, the discovery of iSNVs shared among 44 contacts of a putative index case in a large outbreak of the Delta variant of SARS-CoV-2 (described in Siddie et al *Cell* 2022), was followed by the Centers for Disease Control and Prevention altering their guidance to recommend masking for vaccinated individuals. This has been published since the original submission of this work and is now cited.

While the pathogen and the mode of transmission in this case is clearly quite different, this illustrates how important it is to improve our understanding of the circumstances in which iSNVs are informative, by examining them in controlled experimental conditions as we do in this paper. Further, there are many non-viral pathogens of significant impact that motivate this work. For example, Lee *et al*, *eLife* 2020, used within host diversity to identify a previously undetected super spreader event from a Tuberculosis outbreak. Hall et al, *eLife* 2020, showed that in *Staphylococcus aureus*, transmission between hosts and across body sites was characterized by a wide bottleneck size. More recently, Tonkin-Hill *et al* *Nature Microbiology* 2022, used within host diversity to improve resolution of transmission pairs in *Streptococcus pneumoniae*. They challenged prior assumptions that the bottleneck size of *S. pneumoniae* was 1, or a single cell, which was based on experimental work done by Kono *et al* *Plos Pathogens* 2016. This example emphasizes the importance being open minded on the potential utility of methods like ours, which leverage within host diversity to inform transmission. Such approaches are by no means a panacea for resolving transmission routes, but they have the potential to serve as valuable tools in outbreak investigation.

The reviewer citer Carlson *et al*, *Science* 2014, as evidence that our method would not be applicable

	to HIV. Carlson et al paper was based on single gene amplicons that were subjected to PCR, not unbiased whole genome shotgun sequencing. The methods used by Carlson et al have been superseded by methods that rely on whole genome sequencing to infer bottleneck sizes such as Leonard et al J. Virol 2017 and Ghafari et al J. Virol 2020. Moreover Leitner Curr Opin HIV AIDS 2019, emphasizes to need to account for within host diversity in HIV transmission saying “Phylogenetic reconstruction of HIV transmissions that include within-host HIV diversity have recently been established and made available in several software packages.” These include tools such as PHYLOSCANNER, SCOTTI and QUENTIN among others. Lythgoe, Science, 2021 did an in-depth analysis on the role of within host diversity and concluded that SARS-COV2 was modulated by a tight bottleneck. Their study included 15 household pairs and found that 1 of the 15 pairs shared an iSNV, and also found that other pairs had a discordance where an iSNV was a fixed mutation in another patient. This does suggest that our method would not be very effective in this context, but it is important to also note that despite being household contacts, it is not certain that all 15 pairs were actual transmission pairs, they could have been infected through an unsampled host(s). Additionally, the presence of the shared variant in one pair suggests that there could be scenarios where measuring AF could be helpful in SARS-CoV2 transmission, for example in a super spreader event such as that described in Siddle et al.
Minor points:	
Equations 1 and 2 were not displayed properly in the manuscript I received. For example I think that equation 1 should have $P(T \theta)$ on the right hand side, not simply $P(\theta)$. I think this is just a formatting error?	These were typographic errors that have now been corrected.
Not all of the data shown in Figures 4D-F seems to be appropriate for a box plot. In particular, in Figure 4F it looks as though the ‘false’ data are bimodal: there are so many	The reviewer makes a good point; however, it should be noted that there are up to 22 transmission events in each step. The false category includes pairs that are over 10 steps apart as well as some that are as little as two

outliers that few conclusions can be drawn from what is shown.	transmission steps apart. Thus, the level of heterogeneity highlighted by the reviewer is to be expected. In figure 4A, we showed the density distributions of the mean change in allelic frequency, grouped by number of transmission steps. There you can see that the distribution of mean AF change in proximal pairs (2-5) are shaped more similarly that those that are more distant (10+) transmission pairs. This is not a flaw of our method; it reflects the biology of transmission pairs and the underlying premise that as things transmit further apart, they become more distant. To say that a conclusion cannot be drawn from this is to assume that our method is 100% accurate, which is not what we claim. However, our approach significantly improves the ability to differentiate transmission pairs, and this figure supports that claim.
Line 234: “the bottleneck size is greater than or comparable to the amount of within-host diversity” - Please clarify: in numerical terms the two statistics are measured using different units.	This has been reworded to read: “which suggests that despite being small, the bottleneck size is large enough to accommodate multiple haplotypes in a transmission event”

Reviewer #1' comments:

Here are my thoughts: the authors responded well to many of the concerns and the study clearly has merit, but an important unresolved point remains with respect to in which cases it would be applicable and comparisons to SNP based approaches alone (without iSNV information). There are limited to no experimental comparisons (simulated/real) data that clearly show the strengths/weaknesses of the iSNV based model for transmission inference. As a first step, a detailed simulated analysis similar to Figure 4 in the following paper where the authors compare transmission tree accuracy with (a) SNPs only, (b) iSNVs only, (c) SNPS+iSNVs, would add scientific rigor to this study and scheme (citation 7):

<https://academic.oup.com/aje/article/186/10/1209/3860343> .There they could clarify, for a range of shared iSNV proportions and bottleneck sizes, when and where their approach would be most useful (albeit based on a simulated setting).

But more importantly, there are no comparisons to any existing approaches. If tools like Quentin (<https://pubmed.ncbi.nlm.nih.gov/29304222/>, designed to elucidate transmission networks on iSNVs in a bayesian framework) are not applicable as the authors state, then they should still be cited and they should clarify in a table/introductory text why previous approaches capable of leveraging iSNVs are not appropriate. Along these lines, TransPhylo indicates "TransPhylo can infer the transmission tree from a dated phylogeny in a way that accounts for within-host evolution". The authors could compare their approach on data available from this recent TransPhylo publication (<https://royalsocietypublishing.org/doi/10.1098/rstb.2021.0246>) and compare/contrast transmission chains (a reference to this work is lacking).

Finally, while much older data, the authors missed opportunity to revisit the Klebsiella outbreak data where iSNVs were not leveraged to see if that would increase concordance with the known epidemiological data (this study reported key discordance when using SNP only data).

<https://www.ncbi.nlm.nih.gov/pmc/articles/PMC3521604/>. I highlight this as one such example where the authors could have expanded upon their analysis and compared to previous studies to compare/contrast/interrogate the value of iSNVs + SNPS for transmission chain inference.

In summary, while I have no doubt the manuscript represents a valuable contribution, I do have concerns with respect to the lack of experimental validation & comparison to similar/existing tools (or at minimum, clear & fully justified explanation as to why these tools are fundamentally unable to be used in an evaluation). I am a bit less concerned about applicability to other pathogens, as tools that operate on iSNVs (such as those presented by the authors) are needed and valuable even for a subset of known pathogens.

Reviewer 1 additional comments	Response to reviewer comments
Here are my thoughts: the authors responded well to many of the concerns and the study clearly has merit, but an important unresolved point remains with respect to in which cases it would be applicable and comparisons to SNP based approaches alone (without iSNV information). There are limited to no experimental comparisons (simulated/real) data that clearly show the strengths/weaknesses of the iSNV based model for transmission inference. As a first step, a detailed simulated analysis similar to Figure 4 in the following paper where the authors compare transmission tree accuracy with (a) SNPs only, (b) iSNVs only, (c) SNPs+iSNVs, would add scientific rigor to this study and scheme (citation 7): https://academic.oup.com/aje/article/186/10/1209/3860343 .There they could clarify, for a range of shared iSNV proportions and bottleneck sizes, when and where their approach would me most useful (albeit based on a simulated setting).	The reviewer makes a valid observation on the need to explore the utility of our method over a range of parameters, compared to a SNP-based approach. We have now included a model that simulates the emergence of iSNVs and changes in allelic frequency over successive transmission steps with varying bottleneck sizes in lateral transmission chains (Figure 6). To achieve this, we designed a model that simulated transmission chains while allowing for variations in bottleneck sizes. Our model relied on two main assumptions. Firstly, we assumed that prior to transmission, individual single nucleotide variants (iSNVs) experience selection pressure towards fixation, and the resulting change in allelic frequency depends on the initial allele frequency and a constant coefficient, regardless of bottleneck size. Secondly, we postulated that at the point of transmission, iSNVs undergo a secondary change in allelic frequency due to the bottleneck, which can either drive them towards fixation or towards their elimination. Additionally, we considered stochastic emergence of new iSNVs at a constant rate, as well as the possibility of fixed SNVs reverting back to iSNVs. Initially, we parameterized our model using 50 variable genomic loci, a de novo emergence rate of iSNVs of 0.002 per site per transmission cycle (p), and a selection constant of 3 (S). To further investigate, we explored different combinations of p (0.002, 0.005, 0.01 - representing slow, medium, and fast emergence rates, respectively) and S (1, 3, 5, 10 - representing strong, mild, weak, and very weak selection forces, respectively). While this model is not perfect, it does a good job of showing how changes in bottleneck size can impact the performance of measuring changes in allelic frequency vs SNP distance to infer transmission. It also provides some insight into how rapidly evolving pathogens either via rapid fixation of iSNVs or rapid emergence of de novo iSNVs impacts the performance of both metrics.

But more importantly, there are no comparisons to any existing approaches. If tools like Quentin (https://pubmed.ncbi.nlm.nih.gov/29304222/, designed to elucidate transmission networks on iSNVs in a bayesian framework) are not applicable as the authors state, then they should still be cited and they should clarify in a table/introductory text why previous approaches capable of leveraging iSNVs are not appropriate. Along these lines, TransPhylo indicates "TransPhylo can infer the transmission tree from a dated phylogeny in a way that accounts for within-host evolution". The authors could compare their approach on data available from this recent TransPhylo publication (https://royalsocietypublishing.org/doi/10.1098/rstb.2021.0246) and compare/contrast transmission chains (a reference to this work is lacking).	We have included a paragraph in the introduction that details why these methods are not applicable to our example. Quentin uses Splittree to reconstruct the within host network of a host, it then compares these within host networks and reconstructions transmission. This is very similar to PhyloScanner, which sub-samples a bam file or raw reads and attempts to reconstruct sub-populations in the host. Both these processes introduce bias that may distort the true frequencies of sub-populations. Deep sequencing and measuring allele frequencies does a better job of maintaining the relative frequencies of sub-populations in the host. Transphylo uses a dated phylogeny to infer transmission, to incorporate within host diversity, one simply inputs multiple consensus genomes from the same host, again this relies on consensus sequences and may distort relative proportions of sub-populations. Moreover, it is difficult to generate a time signal from isolates collected over a short time period such as in an emerging outbreak.
Finally, while much older data, the authors missed opportunity to revisit the Klebsiella outbreak data where iSNVs were not leveraged to see if that would increase concordance with the known epidemiological data (this study reported key discordance when using SNP only data). https://www.ncbi.nlm.nih.gov/pmc/articles/PMC3521604/. I highlight this as one such example where the authors could have expanded upon their analysis and compared to previous studies to compare/contrast/interrogate the value of iSNVs + SNPS for transmission chain inference.	We agree with the reviewer, there is an important need to apply our method to existing datasets. However, there are two challenges to finding an appropriate dataset. First, in order to capture within host diversity, the entire sample needs to be collected prior to DNA extraction. This means taking a sweep of all colonies on an agar plate instead of isolating a purified colony. Classically, pure colony isolates are the gold standard in microbiology and most published studies rely on this approach. The second challenge is that the plate sweep must be sequenced to a high sequencing depth in order to minimize the impact of sequencing error and also to capture low frequency variants. In principle the Klebsiella dataset proposed by the reviewer would be an ideal candidate to test our method. It is a well sampled outbreak with patients sampled at multiple times and multiple copies of the index strain have been sequenced. However, the sequencing depth of this study is low,

	according to the supplementary table the average coverage of genomes ranges between 20X and 44X. Moreover, it is not clearly stated that these samples are sequenced from plate sweeps. While the methods do not explicitly detail colony purification steps, it does reference isolation, which can mean colony purification. We previously published a reanalysis of a TB outbreak that was reanalyzed using deep sequencing and showed a previously undetected super spreader event (PMC7012596). This dataset would be ideal as it is deep sequenced and not colony purified, however these data were collected from an indigenous population in Nunavik, Canada. Given the more stringent ethical rules on indigenous populations we would need to resubmit a new round of ethic and IRB in order to reanalyze these data with our method. Nonetheless, we are confident that with time new datasets will be generated that meet the aforementioned criteria and our method can be deployed to aid in inference of transmission chains.
In summary, while I have no doubt the manuscript represents a valuable contribution, I do have concerns with respect to the lack of experimental validation & comparison to similar/existing tools (or at minimum, clear & fully justified explanation as to why these tools are fundamentally unable to be used in an evaluation). I am a bit less concerned about applicability to other pathogens, as tools that operate on iSNVs (such as those presented by the authors) are needed and valuable even for a subset of known pathogens.	The comments made by the reviewer are well received. We have added a paragraph to the introduction as stated above and we have run simulated transmission chains to demonstrate the added value of our method compared to SNP based approaches.

REVIEWERS' COMMENTS

Reviewer #1 (Remarks to the Author):

I've reviewed the manuscript and the authors have adequately addressed my concerns. The additional analyses strengthen the applicability of their findings on this important topic of tracking within host evolution.